# Synthesis of a Two-Dimensional Molybdenum Disulfide Nanosheet and Ultrasensitive Trapping of *Staphylococcus Aureus* for Enhanced Photothermal and Antibacterial Wound-Healing Therapy

**DOI:** 10.3390/nano12111865

**Published:** 2022-05-30

**Authors:** Weiwei Zhang, Zhao Kuang, Ping Song, Wanzhen Li, Lin Gui, Chuchu Tang, Yugui Tao, Fei Ge, Longbao Zhu

**Affiliations:** 1School of Biological and Food Engineering, Anhui Polytechnic University, Wuhu 241000, China; zwwjcf0908@163.com (W.Z.); kz827478890@163.com (Z.K.); songping1987@foxmail.com (P.S.); liwanzhen129@126.com (W.L.); tangcc2333@126.com (C.T.); 2Department of Microbiology and Immunology, Wannan Medical College, Wuhu 241002, China; guilin729@126.com

**Keywords:** MoS_2_ nanosheet, antibacterial activity, photothermal therapy, wound-healing

## Abstract

Photothermal therapy has been widely used in the treatment of bacterial infections. However, the short photothermal effective radius of conventional nano-photothermal agents makes it difficult to achieve effective photothermal antibacterial activity. Therefore, improving composite targeting can significantly inhibit bacterial growth. We inhibited the growth of *Staphylococcus aureus* (*S. aureus*) by using an extremely low concentration of vancomycin (Van) and applied photothermal therapy with molybdenum disulfide (MoS_2_). This simple method used chitosan (CS) to synthesize fluorescein 5(6)-isothiocyanate (FITC)-labeled and Van-loaded MoS_2_-nanosheet hydrogels (MoS_2_-Van-FITC@CS). After modifying the surface, an extremely low concentration of Van could inhibit bacterial growth by trapping bacteria synergistically with the photothermal effects of MoS_2_, while FITC labeled bacteria and chitosan hydrogels promoted wound healing. The results showed that MoS_2_-Van-FITC@CS nanosheets had a thickness of approximately 30 nm, indicating the successful synthesis of the nanosheets. The vitro antibacterial results showed that MoS_2_-Van-FITC with near-infrared irradiation significantly inhibited *S. aureus* growth, reaching an inhibition rate of 94.5% at nanoparticle concentrations of up to 100 µg/mL. Furthermore, MoS_2_-Van-FITC@CS could exert a healing effect on wounds in mice. Our results demonstrate that MoS_2_-Van-FITC@CS is biocompatible and can be used as a wound-healing agent.

## 1. Introduction

Bacterial infections are consistently ranked as one of the leading causes of human mortality, with infection rates, mortality rates, and hospitalization costs increasing annually [1,2,3]. The high prevalence of bacterial infections has led to the misuse of antibiotics, resulting in the emergence of superbugs as bacteria become resistant to treatment. Unfortunately, superbugs are arising at a rate much faster than that of new antibiotic discovery, thereby leading to a growing threat of untreatable bacterial infections [4,5,6,7,8]. Therefore, new treatments are urgently needed.

With the rapid development of technologies in the fields of modern nanotechnology and biomedicine, numerous antibacterial inorganic nanoparticles (NPs), such as silver, gold, copper nanoparticles, alumina, zinc oxide, magnesium oxide, silica titanium dioxide, and graphene oxide NPs, as well as their composites, have been used in antibacterial therapy [3,9,10,11]. For example, two-dimensional (2D) graphene-based nanocomposites and analogues have been demonstrated to have wide application prospects because of their multi-functional antibacterial mechanisms, which include physical and chemical damage to bacterial cells [12,13]. Molybdenum disulfide (MoS_2_), a transition metal dichalcogenide, is similar to graphene. Currently, MoS_2_ has several potential applications in the field of biomedicine due to its unique electronic, optical, mechanical, and chemical properties, and MoS_2_ has been demonstrated to have unique optical properties in that it can convert itself into heat by absorbing light energy when irradiated with near-infrared (NIR) light. Through this property, MoS_2_ can kill bacteria [14,15,16].

Free MoS_2_ without irradiation does not have significant antibacterial effects. To improve the antibacterial ability of MoS_2_, we selected vancomycin (Van). Van is a peptide antibiotic that can kill bacteria, such as *Staphylococcus aureus* (*S. aureus*), residing on a wound’s surface [17,18]. Van has an attractive property in that it can target bacteria through hydrogen bonding to the terminal D-Ala-D-Ala sequence of the cytosolic peptide unit of Gram-positive bacteria. In other words, Van recognizes the D-Ala-D-Ala sequence on the cell surface of bacteria, increasing the half-life and effective working radius of the nano-antibacterial composites and enhancing the antibacterial effect [19,20,21,22]. Together, MoS_2_ and Van can target Gram-positive bacteria, which can further enhance the thermal response of MoS_2_ and promote bacterial growth inhibition while applying antibiotic therapy. Van has a primary amine group that binds covalently to fluorescein 5(6)-isothiocyanate (FITC), and we selected FITC-labeled Van because it can maintain both the fluorescent properties of FITC and the ability of Van to bind to the bacterial cell wall, thereby facilitating subsequent experimental validation.

To promote wound healing, it is important not only to clean the wound of germs but also to maintain an environment that is suitable for healing. A moist and clean environment accelerates the migration of epidermal cells, which facilitates skin cell granulation and division [23,24,25]. Polymer hydrogels are one of the most practiced soft-wet materials used for biomedical applications [26]. They can provide this type of environment for wounds, and thermosensitive hydrogels are easy to prepare. Chitosan (CS) is a biocompatible and weakly immunogenic material that can be degraded by enzymes in vivo, and the degradation products, namely oligosaccharides, are non-toxic. In addition, CS can enhance drug penetration by affecting the tightness between epithelial cells, which makes CS a valuable entity in the biomedical field [27]. As such, we prepared a temperature-sensitive hydrogel excipient by wrapping our synthesized MoS_2_-Van-FITC nanomaterials, which possess photothermic and chemotherapeutic properties, in a hydrogel. By physically mixing CS with a sodium β-glycerophosphate (β-GP) solution, a sol-gel phase transition was achieved at temperatures higher than 37 °C due to the enhanced hydrogen bonding, electrostatic attraction, and hydrophobic interactions between CS and β-GP [28].

In this study (Figure 1), Van not only efficiently inhibited the growth of bacteria on the wound surface but also enhanced the antibacterial effect by targeting bacteria through hydrogen bonding with the terminal D-Ala-D-Ala sequence of the cytosolic peptide unit of Gram-positive bacteria, increasing the half-life of the composite and the effective working radius of the photothermal treatment. Therefore, when MoS_2_ is combined with Van, the effective working radius of MoS_2_ PTT can be further increased to better inhibit bacterial growth while applying anti-infective therapy. The covalent binding of primary amine groups using FITC with MoS_2_-Van confers the nanocomposite to maintain the fluorescent properties of FITC, which can be observed for real-time detection. Additionally, photothermal treatment combined with chemotherapy cleaned the wound surface of pathogenic bacteria, thereby accelerating wound healing.

## 2. Materials and Methods

### 2.1. Materials

Raw MoS_2_ (99.5%), FITC (90%, mixture of 5- and 6-isomers), CS (≥99.8%), and Van HCL were purchased from Aladdin Chemical Reagent Co., Ltd. (Shanghai, China). The beef extract peptone agar medium, *Staphylococcus aureus* ATCC 25,923 cells (*S. aureus*, Gram-positive), and *Escherichia coli* ATCC 11,303 cells (*E. coli*, Gram-negative) were obtained from Anhui Polytechnic University.

### 2.2. Synthesis of MoS_2_ Nanosheets

MoS_2_ nanosheets were synthesized by liquid ultrasonic stripping [29,30,31,32]. In brief, 0.5 g of MoS_2_ was combined with 50 mL of 45% ethanol to form the dispersion system of MoS_2_ (concentration, 10 μg/mL), which was sonicated in a water bath for 12 h. The solution was centrifuged at 12,000 rpm for 15 min, and the supernatant was collected and processed by a rotary evaporator to yield a thin film. The film was weighed, and MoS_2_ was resuspended in deionized water to form the MoS_2_ nanosheet aqueous solution.

### 2.3. Synthesis of MoS_2_-Van-FITC

Van is a heptapeptide-containing glycopeptide antibiotic with a primary amine moiety that binds covalently to FITC [33]. In brief, 4 mL of the MoS_2_ nanosheet aqueous solution (concentration, 0.002 g/mL) was aliquoted, 0.2 mg of Van was added to the solution, and the volume was adjusted to 20 mL. The solution was mixed on a magnetic stirrer at 250 rpm for 6 h and centrifuged at 12,000 rpm for 15 min. The precipitate was resuspended in 20 mL of pure water, and 1 mL of FITC was added. The solution was further mixed for 12 h. MoS_2_-Van-FITC was obtained by centrifugation.

### 2.4. Synthesis of the MoS_2_-Van-FITC@CS Hydrogel

To prepare the hydrogel, 300 mg of CS was added to 18 mL of 0.1 mol HCl solution and mixed on a magnetic stirrer until the CS solution was clarified. Thereafter, 2 mg of sodium β-glycerophosphate was added to 2 mL of 0.001 g/mL MoS_2_-Van-FITC solution, dissolved completely, and mixed for 4 h [27,34,35]. The MoS_2_-Van-FITC@CS hydrogel was obtained by heating in a water bath at 37 °C for 1 h.

### 2.5. Characterization

Ultraviolet-visible (UV–vis) absorption spectra were recorded by UV–vis spectroscopy (model S-3100, Scinco Co., Daejeon, Korea). Fluorescence spectra of MoS_2_-Van-FITC NPs were recorded by fluorescence spectrophotometry (model RF-5301PC, Shimadzu Corp., Kyoto, Japan). Fourier transform infrared (FT-IR) spectroscopy was performed with an FT-IR spectrometer (model Nicolette is50, Thermo Fisher Scientific, Waltham, MA, USA). The ultrastructural characteristics of synthesized MoS_2_ were observed via scanning electron microscopy (SEM; model S-4800, Hitachi, Tokyo, Japan). The Brookhaven Zeta Pals instrument was used to obtain zeta potential measurements and to characterize the optical properties of MoS_2_-Van-FITC (Brookhaven Instruments Corp., Holtsville, NY, USA). NPs of different concentrations were illuminated by using NIR irradiation at 808 nm at different power densities for 15 min, and the temperature was detected with an infrared camera with an accuracy of 0.1 °C [36,37].

### 2.6. In Vitro Antibacterial Assays

The minimum inhibitory concentration (MIC) of MoS_2_-Van-FITC NPs was determined using the 96-well microtitration plate dilution method. In brief, 100 μL of LB medium was added to each well of a sterile 96-well plate, and 100 μL of the drug solution was added to the first well, mixed, and diluted in multiples until the last well was mixed. Thereafter, 100 μL of the mixture was discarded, followed by the addition of 100 μL of the bacterial diluent to each well (final density of bacteria, 1 × 10^6^ CFU/mL). The positive controls were kanamycin and ampicillin. The cells of the NIR group were NIR irradiated (1.5 W/cm^2^) for 6 min and cultured at 37 °C for 12 h. The OD value was measured with a microplate reader [38]. The three independent measurements were averaged, and each treatment group had three wells.

Log-phase *S. aureus* cultures were inoculated (1:40) into medium containing MoS_2_-Van-FITC (100 μg/mL) or MoS_2_ (100 μg/mL) and incubated in a shaking incubator at 37 °C for 12 h. The irradiation power densities were 0.5 W/cm^2^, 1 W/cm^2^, and 1.5 W/cm^2^, and the irradiation times were 0 s, 150 s, and 300 s. Thereafter, the cells were coated on solid medium. The number of live bacteria was calculated using the CFU counting method.

### 2.7. Cellular Uptake Assays

To confirm the effect of Van in capturing *S. aureus* cells, we performed cellular uptake assays. Log-phase *S. aureus* cultures were incubated with different concentrations (15, 30, 45 µg/mL) of MoS_2_-Van-FITC NPs for 12 h. The cells treated with PBS served as the control. *S. aureus* cells were harvested and stained with 4ʹ-6-diamidino-2-phenylindole (DAPI; concentration, 5 µg/mL) for 15 min. Thereafter, the cells were washed twice with PBS, and the red and blue channels were examined under a fluorescence microscope.

### 2.8. LIVE–DEAD Assays

To further investigate the antibacterial ability of the drug, we conducted LIVE–DEAD assays. Log-phase *S. aureus* cultures were inoculated (1:40) in medium containing MoS_2_-Van-FITC (100 μg/mL), MoS_2_ (100 μg/mL), or Van (1 μg/mL) and incubated in a shaking incubator at 37°C for 12 h. The irradiation power density was 1.5 W/cm^2^, and the irradiation time was 300 s. Thereafter, the cells were stained with SYTO9 and propidium iodide for 30 min in the dark, washed twice with PBS, and red and green channels were examined under a fluorescence microscope. The number of non-viable bacterial cells was determined by the CytoFLEX system (Beckman Coulter, Brea, CA, USA) [2,39].

### 2.9. Cell Integrity Assays

Log-phase *S. aureus* cultures were treated with different concentrations (25, 50, 100 µg/mL) of MoS_2_-Van-FITC NPs. Van (1 µg/mL) and MoS_2_ (100 µg/mL) served as the controls. The PBS-treated group served as the blank. The irradiation power density was 1.5 W/cm^2^, and the irradiation time was 6 min. The cells were collected by centrifugation at 4000 rpm for 10 min, washed twice with PBS, and fixed with 2.5% glutaraldehyde at 4 °C for 12 h. The cells were subjected to gradient dehydration with different concentrations of ethanol (35%, 50%, 70%, 80%, 95%, 100%) for 20 min each time and then placed into acetone [40,41]. The specimens were observed under a scanning electron microscope.

### 2.10. Establishment of the Wound Mice Model

Kunming female mice (~25 g body weight; 4–5 weeks old) were obtained from the Model Animal Research Center of Nanjing University (Nanjing, China) and housed under standard environmental conditions (temperature, 22 ± 3 °C; humidity, 55 ± 5%; 12-h dark/12-h light cycles). All animals were housed according to the guidelines in the “Guide for the Care and Use of Laboratory Animals”. All animal studies were approved by Suzhou University (Suzhou, China) (approval number: 202010A415). After 7 days of acclimatization, an oval wound of approximately 1.5 cm in length was made by shaving the back of each mouse under anesthesia and adding 100 μL of activated *S. aureus* cells (1 × 10^6^ CFU/mL) dropwise to each wound for two consecutive days. The wound was treated after inflammation [14,42].

### 2.11. Wound Healing Assays

The mice whose wounds were infected with *S. aureus* were randomly divided into five groups (*n* = 5–7 mice) as follows: blank group, MoS_2_ group, MoS_2_-Van-FITC@CS group, NIR MoS_2_ group, and NIR MoS_2_-Van-FITC@CS group. The wound site of the blank group was treated with 100 μL of PBS, that of the MoS_2_ group was treated with 100 μL of MoS_2_ each day, that of the MoS2-Van-FITC@CS group was treated with 100 μL of MoS_2_-Van-FITC@CS hydrogel each day, that of the NIR MoS_2_ group was treated with MoS_2_, followed by irradiation at 1.5 W/cm^2^ for 5 min, and that of the NIR MoS_2_-Van-FITC@CS group was treated with MoS_2_-Van-FITC@CS, followed by irradiation at 1.5 W/cm^2^ for 5 min. The mice were photographed daily for eight consecutive days. The mice were euthanized, and their epidermises were harvested for hematoxylin–eosin staining [43].

### 2.12. Safety Evaluation of MoS_2_-Van-FITC@CS

To examine the toxic effects of NIR and MoS_2_-Van-FITC@CS on the heart, liver, spleen, lungs, and kidneys of mice, the wounded mice were divided into four groups as follows: control group, NIR irradiation group, MoS_2_-Van-FITC@CS group, and MoS_2_-Van-FITC@CS + NIR group. The irradiation power density was 1.5 W/cm^2^, the wavelength was 808 nm, and the irradiation time was 6 min. The mice were treated until their wounds healed, and they were euthanized 30 days after the end of treatment. Their organs were collected, and the cross-sections were stained with hematoxylin–eosin [44].

## 3. Results and Discussion

### 3.1. Characterization of MoS_2_-Van-FITC@CS

The nanocarriers were exfoliated by liquid phase ultrasound. The composite nanomaterials conjugated to FITC-labeled Van showed strong antibacterial effects. After irradiation, MoS_2_ converted light energy into heat energy, thereby killing the bacteria by actively trapping the cells through Van. MoS_2_-Van-FITC had good antibacterial and wound-promoting abilities through the temperature-sensitive hydrogel formed with chitosan (Figure 2A). To determine the synthesis of two-dimensional MoS_2_ nanosheets, the morphology and thickness of MoS_2_ NPs were observed via SEM (Figure 2B). The results of SEM showed that the flake NPs, which had a thickness of approximately 40 nm, were uniformly distributed. Zeta potentiometry can be used to determine the solid–liquid interfacial electrical properties of dispersed systems of particulate matter, so we can determine the successful synthesis of materials by using the potential changes of nanomaterials. The zeta potential of MoS_2_ was −21.4 ± 1.2 mV, whereas the loading of Van resulted in a potential of −8.3 ± 1.9 mV and a zeta potential of −33 ± 0.8 mV after labeling with FITC (Figure 2C). The synthesis of MoS_2_-Van-FITC@CS causes changes in the structure of a single component as a result of electron leaps between electronic energy levels in the valence and molecular orbitals, which can be observed by UV-vis spectrum (Figure 2D). In the UV-vis spectrum, MoS_2_ NPs were observed to have an absorption peak near 808 nm, showing a longitudinal surface plasmon resonance band, which indicated photothermal effects, whereas Van did not show an absorption peak near 808 nm. Van was adsorbed on MoS_2_, after which this characteristic peak significantly shifted. After FITC was decorated on the Van surface, the characteristic peak of MoS_2_-Van-FITC was significantly shifted, and a new characteristic peak was observed near 450 nm. MoS_2_-Van-FITC@CS also showed a shift in the characteristic peak compared to MoS_2_. FTIR spectroscopy allows the observation of the functional groups and chemical bonds contained in the material, in order to be able to determine the successful synthesis of MoS_2_-Van-FITC nanocomposites. FT-IR spectral analysis showed that the presence of Van resulted in an amino peak near 3000 cm^−1^, and a distinct peak at 1000 cm^−1^ in the fingerprint region after the loading of FITC (Figure 2E). A significant change was also found in the fingerprint region after the wrapping of chitosan. The fluorescence properties of MoS_2_-Van-FITC@CS are shown in Figure 2F,G. MoS_2_-Van-FITC@CS emitted fluorescence under UV light irradiation at 465 nm, and the fluorescence properties of the nanomaterials were further confirmed by the fluorescence spectra. Taken collectively, these findings indicate that MoS_2_, which has photothermal properties, was successfully synthesized, Van was successfully loaded, and the surface was modified by FITC. The morphological features of the MoS_2_-Van-FITC@CS hydrogel were indicative of its successful synthesis.

### 3.2. In Vitro Photothermal Efficiency

MoS_2_ is a photothermal agent that produces a large amount of heat to kill bacteria in the NIR region of 808 nm [45,46]. Therefore, we measured its photothermal conversion efficiency to understand the photothermal properties of MoS_2_. As expected, MoS_2_-Van-FITC acted as a good photothermal nanomaterial under NIR irradiation in that it converted light energy into heat energy, resulting in rapid warming (Figure 3A,B). Furthermore, the MoS_2_-Van-FITC concentration was 400 µg/mL, m_PBS_ (m_D_) was 1.0 g, C_H_2_O_ (C_D_) was 4.2 J/g/°C, ΔT_max_ was 47.3 °C (Figure 3C), I was 2 W, and $\tau_{s}$ was 296 s (Figure 3D). Thus, the photothermal conversion efficiency (η) of MoS_2_-Van-FITC was 52%. Appendix A shows the thermal images of MoS_2_-Van-FITC at different concentrations under an NIR irradiation at 808 nm. In summary, MoS_2_-Van-FITC has good photothermal conversion efficiency and can be used as a photothermal nanomaterial for killing bacteria.

### 3.3. In Vitro Antibacterial Activity

We used *S. aureus* and *E. coli* as Gram-positive and Gram-negative bacteria, respectively, in subsequent experiments (Table 1). In the MIC test, we found that MoS_2_-Van-FITC + NIR had the highest killing effect against *S. aureus*, which may have been related to the fact that Van is a narrow-spectrum antibiotic that is only effective against Gram-positive bacteria. MoS_2_-Van-FITC + NIR showed the effects of common antibiotics at low doses, and the inhibition of growth made it difficult for bacteria to develop resistance (Table 1). Thermal images of the test in MIC were showed in Appendix A. Figure 4 shows the thermogram of the solution temperature increase after NIR irradiation. We found that our nanomaterials had a stronger growth inhibition ability against *S. aureus*. In the CFU test, we screened the power density and light time using power densities of 0.5, 1, and 1.5 W/cm^2^, and light times of 0, 150, and 300 s. We observed that a power density of 1.5 W/cm^2^ and a light time of 300 s inhibited bacterial growth. As shown in Figure 4, the inhibition rate of bacteria in the absence of MoS_2_ was low, and the survival rate of bacteria was 89%. However, the survival rate of bacteria gradually decreased after NIR irradiation, and the survival rate of bacteria was only 4.2% after treatment with MoS_2_ (100 μg/mL, 1.5 W/cm^2^, 300 s). After treatment with MoS_2_-Van-FITC + NIR (100 μg/mL), the survival rate of bacteria was 0.9% after an irradiation time of 300 s at a power density of 1.5 W/cm^2^. In the in vitro antibacterial test, the nanomaterials inhibited the growth of *S. aureus* cells, which played a role in the elimination of bacteria from the wound, thereby speeding up wound healing.

### 3.4. Cellular Uptake Assays

Van targets Gram-positive bacteria by binding to the hydrogen bond of the terminal D-Ala-D-Ala sequence of the cytosolic peptide of bacteria. It is also a heptapeptide-containing glycopeptide antibiotic with a primary amine moiety that binds covalently to FITC [46,47,48]. Van can label FITC on the bacterial surface; therefore, we verified the targeting of Van by DAPI staining. When MoS_2_-Van-FITC was used at a concentration of 15 μg/mL, most of the bacteria were labeled, similar to higher concentrations of 30 μg/mL and 45 μg/mL. However, as the concentration increased, the number of bacteria decreased, showing the excellent antibacterial ability of MoS_2_-Van-FITC (Figure 5). The results of cellular uptake assays indicated that MoS_2_-Van-FITC successfully targeted bacteria and showed excellent antibacterial ability, thereby achieving our goal of using combined chemotherapy and photothermal therapy to inhibit bacterial growth.

### 3.5. Fluorescent Staining Analysis of Antibacterial Activity

The CFU assay can detect only viable bacteria to determine the antibacterial activity of nanomaterials. To further examine the antibacterial activity of nanomaterials, we determined the number of viable and non-viable cells using the LIVE–DEAD assay to examine the antibacterial activity of MoS_2_-Van NPs. Viable bacterial cells were stained green, whereas non-viable bacterial cells were stained red (Figure 6A). No cell death was observed in the blank group. However, cell death was observed in the Van group, indicating that Van can inhibit bacterial growth. Similar results were obtained for the cells treated with MoS_2_ + NIR and MoS_2_-Van, with the MoS_2_-Van + NIR group exhibiting stronger inhibition of bacterial growth after NIR irradiation.

To further confirm that MoS_2_-Van+NIR reduced the survival of bacteria, the number of viable and non-viable bacterial cells was quantified via flow cytometry. As shown in Figure 6B, the apoptotic rate of the blank + NIR group was 0.78%. The apoptotic rate of MoS_2_ + NIR (100 µg/mL) after NIR irradiation was 53.95%. When the concentration of Van was 1 µg/mL, the apoptotic rate was 49.68%. The apoptotic rate of MoS_2_-Van (100 µg/mL) was 67.24% without irradiation, which was mainly due to the effects of Van, but MoS_2_ also played its own role after NIR irradiation, and the apoptotic rate was 94.51%. The increased antibacterial activity of MoS_2_-Van NPs was further confirmed by the quantitative analysis of viable and non-viable cells via flow cytometry.

### 3.6. Cell Integrity Study

Based on our findings, MoS_2_-Van NPs + NIR showed efficient antibacterial activity against *S. aureus*. Because photothermal action mainly targets the bacterial cell surface, and the cell surface is also the site of action of Van, we speculate that changes in cell integrity may be the main mechanism behind the induction of apoptosis in bacteria. As such, we investigated the effects of MoS_2_-Van NPs + NIR on the cellular integrity of *S. aureus* by SEM.

The integrity of bacterial cells was examined via SEM, as shown in Figure 7. *S. aureus* cells in the blank group had normal cell morphology, including intact cell membranes. The results showed that NIR irradiation alone did not affect the structure of cells. The rupture and shrinkage of cells could be clearly seen after treatment with Van and MoS_2_ in the control group, and the enlarged area showed that the cells did not have intact cell membranes. In addition, there was cell leakage. In the MoS_2_-Van-FITC group, we observed more severe cell damage, even at a concentration of 25 µg/mL, compared to the blank group. As the concentration increased, the cell damage increased, indicating that MoS_2_-Van-FITC had a stronger antibacterial effect. The results from the elemental analysis chart showed that the bacterial surface did contain elemental sulfur and molybdenum, indicating that the bacterial surface contained MoS_2_. Taken collectively, these findings indicate that MoS_2_-Van-FITC NPs have very effective antibacterial activity compared to MoS_2_ NPs and Van alone.

### 3.7. In Vivo Wound Healing Evaluation

We established a wound-healing mouse model and examined the pro-wound healing effects by directly applying NPs combined with irradiation. As shown in Figure 8, the MoS_2_-Van-FITC@CS hydrogel alone was slightly therapeutic, and the rate of wound healing increased with irradiation (Figure 8A). Heating, which increased the temperature of the nanomaterial to 50 °C by irradiation at 1.5 W/cm^2^ for 6 min, achieved a good therapeutic effect. The MoS_2_ group had the largest relative wound area, which did not heal. However, MoS_2_ decreased the relative wound area after NIR irradiation. MoS_2_-Van-FITC@CS hydrogel + NIR had the best therapeutic effect after NIR irradiation, where the relative wound area was reduced to 20.8%. The weight of mice in all groups decreased and then increased. The reason for the decrease in weight was likely due to the appearance of wounds, but as treatment progressed, the mice recovered and regained the weight. MoS_2_ was least effective, and the healing of mice treated with MoS_2_ was similar to that of the controls (Figure 8B). Immunohistochemical analysis was performed to examine the epidermis from the different groups of mice, and the abnormal histological features could be clearly seen in the stained sections (Figure 8D). No changes in morphology were observed in the epidermises of mice in the control group and the other four groups. The epidermises showed no significant damage compared to those of normal mice, indicating that the elevated temperature of the nanosheets during the photothermal treatment did not cause significant damage to the wounds. Therefore, the therapeutic effect of the MoS_2_-Van-FITC@CS hydrogel combined with NIR irradiation showed that there was no harm to the mice and their wound healing was accelerated. The preliminary photothermal imaging of mice also demonstrated the thermal response of NPs. In summary, the MoS_2_-Van-FITC@CS hydrogel combined with photothermolysis significantly promoted wound healing in mice.

### 3.8. Biosafety Evaluation

The evaluation of toxicity is important for each new drug. As such, we evaluated the safety of our hydrogels in vivo. We applied the hydrogels to the wounds of healthy mice, followed by NIR irradiation, and performed a routine histological analysis of the major organs (Figure 9). Compared with the controls, no significant organ damage, tissue edema, cell death, or inflammatory cell infiltration was observed in the examined organs after treatment with NIR irradiation, MoS_2_-Van-FITC@CS, and MoS_2_-Van-FITC@CS + NIR, indicating that MoS_2_-Van-FITC@CS + NIR was non-toxic in mice.

## 4. Conclusions

In this study, the Van-modified MoS_2_-loaded nanosystem, which was encapsulated in a chitosan hydrogel, was established to examine its antibacterial activity and wound healing ability. The results showed that the thickness of MoS_2_ NPs was <100 nm, whereas other experiments revealed that the surface of MoS_2_ was successfully modified by Van. The antimicrobial activity was enhanced when Van was labeled. The photothermal characterization experiments confirmed that MoS_2_ had good photothermal conversion efficiency, and cellular uptake assays verified the active capture of *S. aureus* by Van, which significantly improved the photothermal inhibition of bacterial growth. The results of in vitro experiments indicated that NIR could significantly increase the antibacterial activity of MoS_2_ NPs, whereas those of flow cytometry showed that NPs could increase the apoptotic rate of bacterial cells. The morphological features of bacterial cells treated with MoS_2_-Van-FITC NPs and NIR irradiation were examined, and NPs were observed to disrupt the integrity of the bacterial cell wall. Furthermore, MoS_2_-Van-FITC combined with NIR irradiation could disrupt the cell morphology, induce apoptosis, and affect cell proliferation in vitro. In a wound healing assay, the MoS_2_-Van-FITC@CS hydrogel could accelerate wound healing. In summary, MoS_2_ in combination with NIR irradiation shows good applicability in the inhibition of bacterial growth, and the CS hydrogel in combination with a photothermal agent that actively traps *S. aureus* can disinfect the wound and maintain a moist environment to accelerate wound healing.

## Figures and Tables

**Figure 1 nanomaterials-12-01865-f001:**
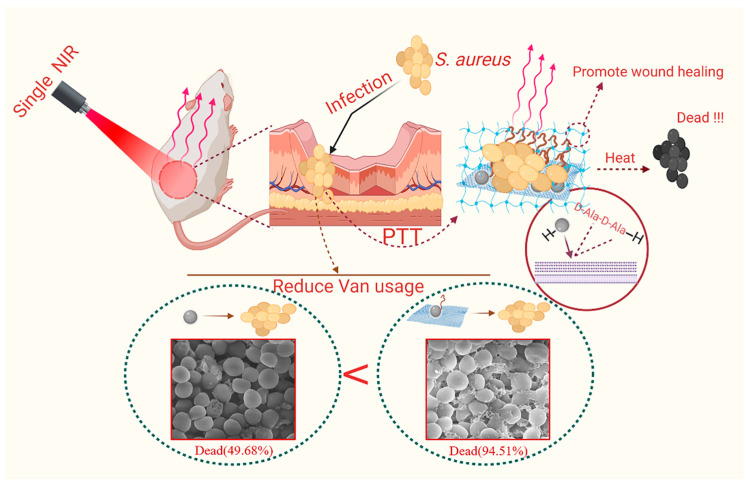
Schematic illustration of the synergism between the two components of MoS_2_ and Van in MoS_2_-Van-FITC@CS, which was constructed and applied to wound healing in vivo with their highly efficiently antibacterial properties.

**Figure 2 nanomaterials-12-01865-f002:**
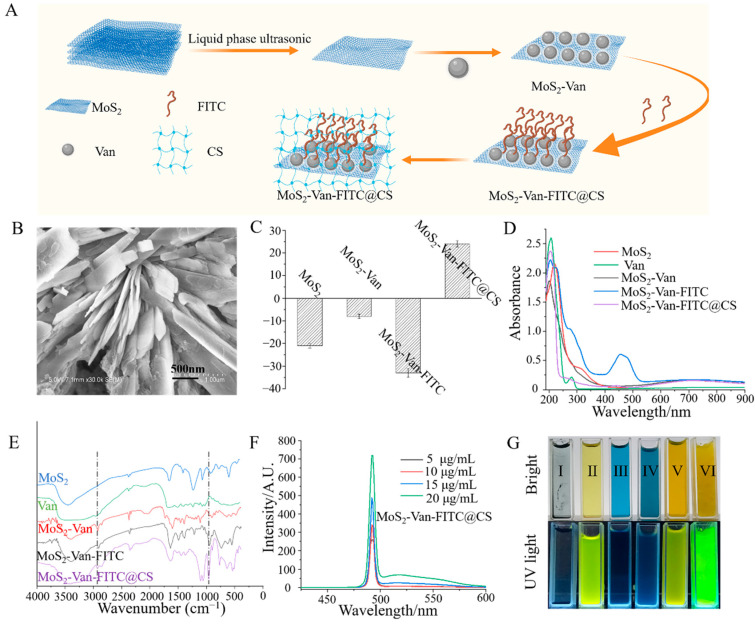
Synthesis and characterization of MoS_2_-Van-FITC@CS. (**A**) Synthesis illustration of MoS_2_-Van-FITC@CS. (**B**) SEM images of MoS_2_ NPs. (**C**) Zeta potential of MoS_2_, MoS_2_-Van, MoS_2_-Van-FITC, and MoS_2_-Van-FITC@CS. (**D**) UV-vis absorption spectra of MoS_2_, Van, MoS_2_-Van, MoS_2_-Van-FITC, and MoS_2_-Van-FITC@CS. (**E**) FT-IR spectrometry of MoS_2_, Van, MoS_2_-Van, MoS_2_-Van-FITC, and MoS_2_-Van-FITC@CS. (**F**) Digital images of I (Milli-water), II (FITC), III (MoS_2_), IV (MoS_2_-Van), V (MoS_2_-Van-FITC), and VI (MoS_2_-Van-FITC@CS) under bright and UV light. (**G**) Fluorescence emission spectra of MoS_2_-Van-FITC@CS.

**Figure 3 nanomaterials-12-01865-f003:**
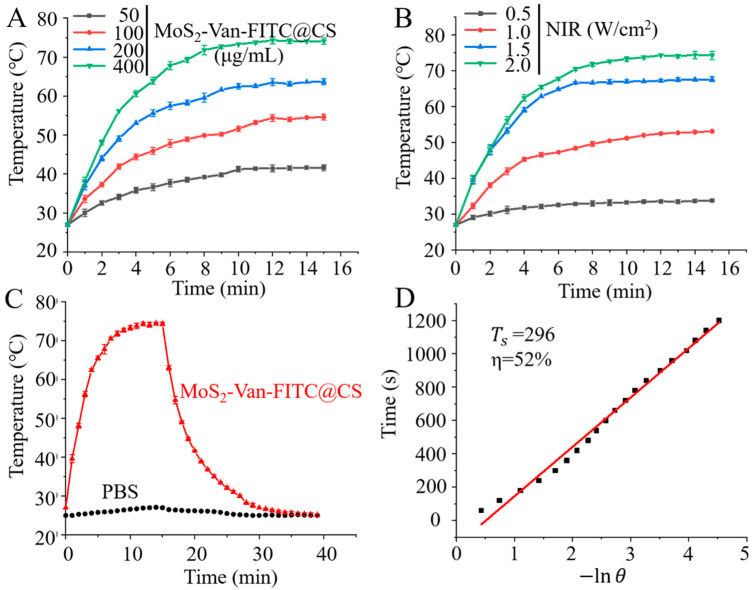
In vitro photothermal efficiency. (**A**) The photothermal responses of MoS_2_-Van-FITC@CS. Different concentrations (50, 100, 200 and 400 µg/mL) were exposed to NIR irradiation (808 nm; 2 W/cm^2^). PBS served as a control. (**B**) The photothermal responses of MoS_2_-Van-FITC@CS (400 µg/mL), which was exposed to different power density of NIR irradiation (808 nm; 0.5, 1, 1.5 and 2 W/cm^2^). (**C**) The heating and cooling curves of MoS_2_-Van-FITC@CS (400 μg/mL) and PBS (laser irradiation at 808 nm). (**D**) The linear regression between the cooling period and −ln(*θ*) of the driving force temperature. Results shown are mean ± SD, *n* = 3.

**Figure 4 nanomaterials-12-01865-f004:**
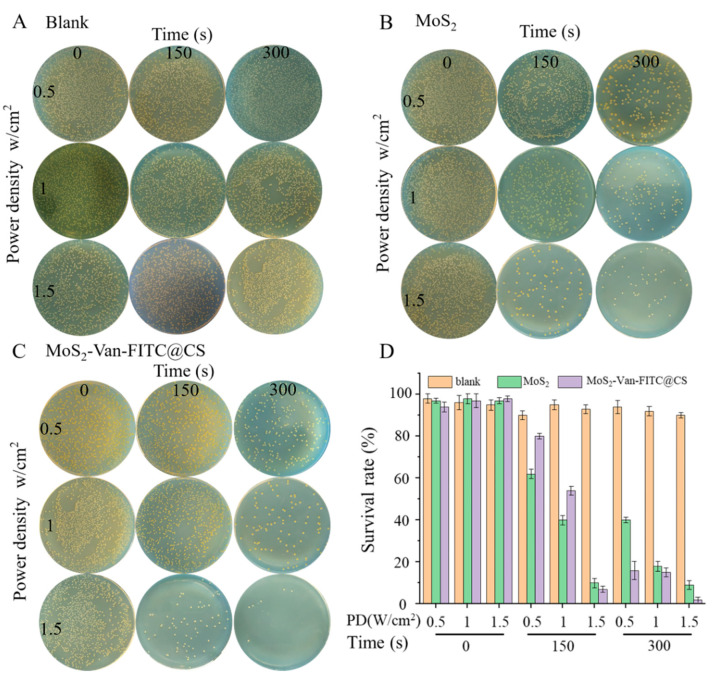
In vitro antibacterial activity. (**A**) The results of the CFU assay for the blank group. PBS as a blank group. (**B**) The results of the CFU assay for MoS_2_ (100 μg/mL) for different times (0, 150 and 300 s) at different power densities (0.5, 1 and 1.5 W/cm^2^). (**C**) The results of the CFU assay for MoS_2_-Van-FITC (100 μg/mL) for different times (0, 150 and 300 s) at different power densities (0.5, 1 and 1.5 W/cm^2^). (**D**) Quantitative statistical results of (**A**–**C**) (PD = Power Density). graphed using the Origin software. Results shown are mean ± SD, *n* = 3.

**Figure 5 nanomaterials-12-01865-f005:**
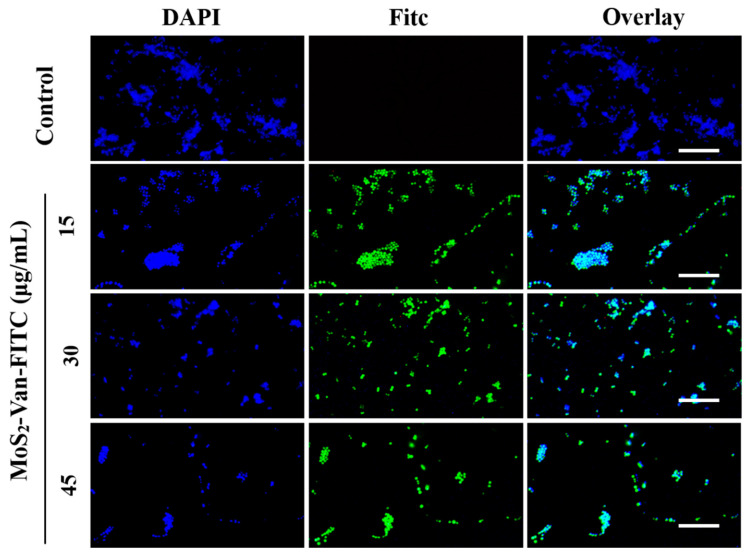
Fluorescence microscopy images of *S. aureus* cultures treated with MoS_2_-Van-FITC at various concentrations (15, 30, 45 µg/mL) for 12 h. PBS served as a control for the blank group. The cells stained with DAPI for 30 min were fluorescent blue, and those stained with MoS_2_-Van-FITC were fluorescent green. Scale bar = 15 μm. (DAPI, Ex = 358 nm and Em = 461 nm; FITC, Ex = 490 nm and Em = 525 nm).

**Figure 6 nanomaterials-12-01865-f006:**
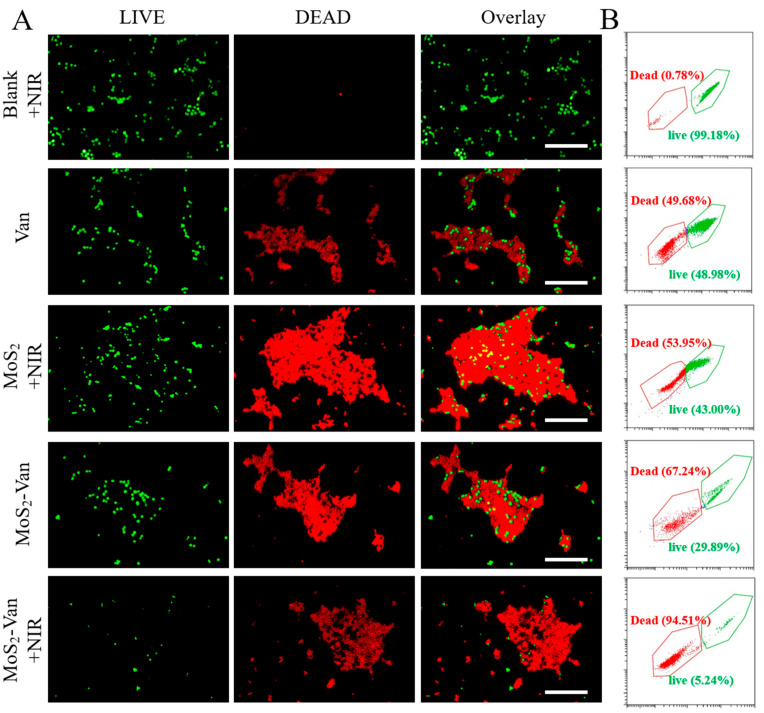
Confocal fluorescence microscopy assay (**A**). *S. aureus* cultures after treatment with MoS_2_-Van NPs + NIR (100 µg/mL). Van solution (1 µg/mL), MoS_2_ NPs + NIR (100 µg/mL) and MoS_2_-Van (100 µg/mL) served as control groups. PBS served as a blank group. The cells were stained with SYTO 9 (green fluorescence) and PI (red fluorescence) for 30 min. The cells underwent NIR irradiation at 808 nm (1.5 W/cm^2^, 6 min). The results of the apoptotic assay by flow cytometry analysis were statistically analyzed by CytExpert software (version 2.4.0.28) (**B**). Scale bar = 15 μm. The data are expressed as mean ± SD (*n* = 3).

**Figure 7 nanomaterials-12-01865-f007:**
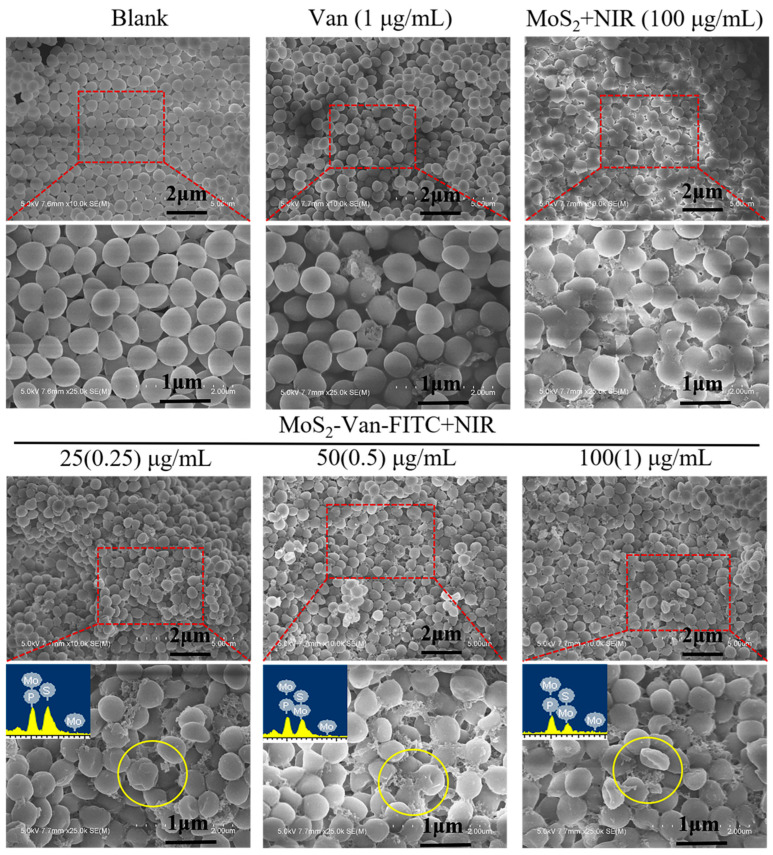
SEM images of *S. aureus* cells. The bacterial cultures were treated with MoS_2_-Van-FITC NPs at various concentrations (25, 50 and 100 µg/mL). The bacterial cultures treated with Van (1 µg/mL) and MoS_2_ NPs + NIR (100 µg/mL) served as control groups. PBS + NIR served as a blank group. The red squares indicate the enlarged regions. The blue area is the elemental analysis chart. The cells underwent NIR irradiation at 808 nm (1.5 W/cm^2^, 6 min).

**Figure 8 nanomaterials-12-01865-f008:**
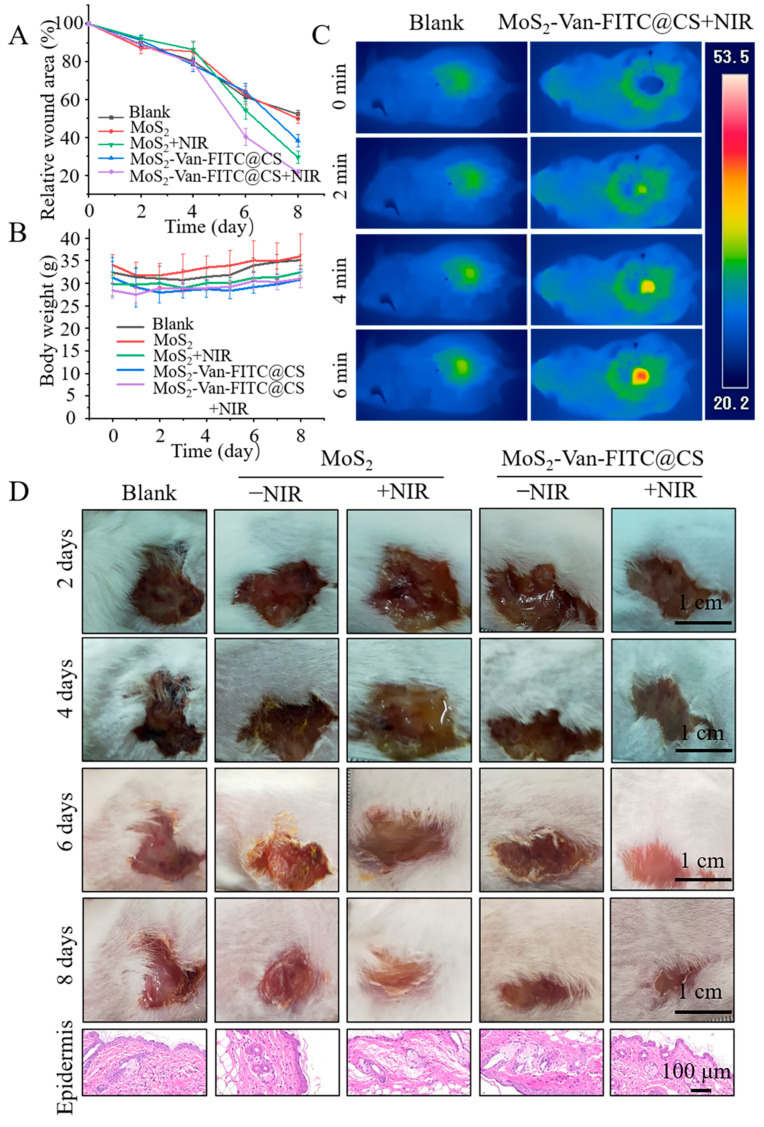
In vivo wound healing analysis. (**A**) The relative wound area curve shows the MoS_2_-Van-FITC@CS hydrogel had a significant positive effect on wound healing. (**B**) The body weights of different groups after treatment. (**C**) The thermal imaging of mice after treatment. The cells underwent NIR irradiation at 808 nm (1.5 W/cm^2^, 6 min). (**D**) The photographs of wounds were taken every 2 days after treatment. MoS_2_-Van-FITC@CS hydrogel + NIR (100 µg/mL), MoS_2_ NPs (100 µg/mL), MoS_2_ NPs + NIR (100 µg/mL) and MoS_2_-Van-FITC@CS hydrogel served as control groups. PBS + NIR served as a blank group. The cells underwent NIR irradiation at 808 nm (1.5 W/cm^2^, 6 min). Scale bar = 1 cm. Results shown are mean ± SD, *n* = 5–7.

**Figure 9 nanomaterials-12-01865-f009:**
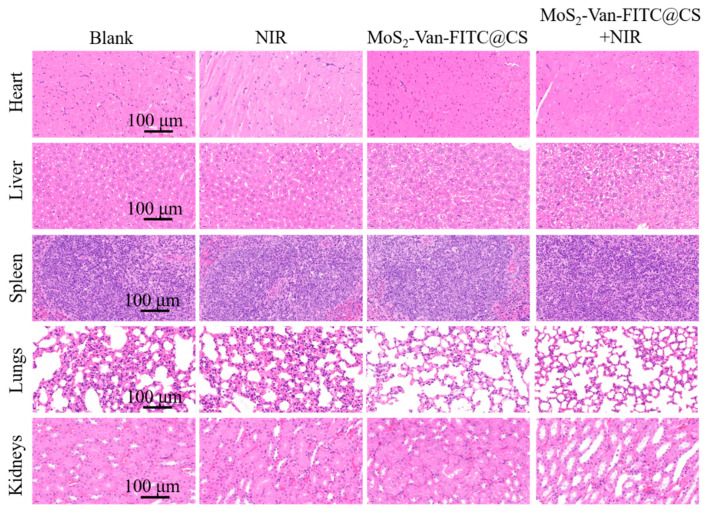
In vivo toxicity evaluation. The hematoxylin–eosin-stained images of major organs following different treatments of normal mice. Group 1 was the no treatment group, group 2 was the NIR irradiation (1.5 W/cm^2^, 6 min) group, group 3 was the MoS_2_-Van-FITC@CS group and group 4 was the MoS_2_-Van-FITC@CS + NIR (1.5 W/cm^2^, 6 min) group.

**Table 1 nanomaterials-12-01865-t001:** Antibacterial activities with MICs values test (μg/mL).

Materials	Bacteria	
	*S. aureus*	*E. coli*
MoS_2_	>128	>128
Van	2	>128
MoS_2_-Van-FITC	64	64
MoS_2_-Van-FITC@CS	64	64
MoS_2_ + NIR	64	128
MoS_2_-Van-FITC + NIR	36	128
MoS_2_-Van-FITC@CS + NIR	36	128
Kanamycin	6	12
Ampicillin	18	24

Data are average values of at least three replicates.

## Data Availability

Not applicable.

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
