# Peer review of "Synthesis of a Two-Dimensional Molybdenum Disulfide Nanosheet and Ultrasensitive Trapping of Staphylococcus Aureus for Enhanced Photothermal and Antibacterial Wound-Healing Therapy"

_nanomaterials, 2022, doi:10.3390/nano12111865_

Round 1
Reviewer 1 Report
Dear authors,
I have attached a document containing my main comments about your paper.
Best regards

Author Response
Dear Reviewer.
Thank you for your valuable comments. I have enclosed a document containing the changes I have made in response to your comments.
Best regards

Reviewer 2 Report
Dear Author,
I find this manuscript acceptable for publication in Nanomaterials after minor revision. Some parts are not well organized, and minor inconsistencies must be corrected to improve the quality of the article.
Comments:
1. Synthesis of MoS2-Van-FITC: i) The author refers to reference 30, where just bacteria staining with FITC is described; ii) Did authors observe unspecific adsorption of FITC to MoS2 sheet without Van? Why was Van-FITC conjugate not synthesized without MoS2 sheet?
2. In vitro antibacterial assay: Did the authors study the MIC of MoS2-Van-FITC without FITC labeling? Could authors show or discuss the possible effect of FITC label on the antibacterial properties of the synthesized complex? Further, the systematic design of the experiment should be maintained (e.g. testing of time and intensity of the irradiation then MIC…)
3. I would recommend reorganizing the subchapter 3.1. and perhaps to separate Fig. 2A from graphs. Also, the beginning of the paragraph sounds confusing. The author should start with the characterization of NPs with SEM and zeta potential and continue with UV-VIS, FTIR etc. Further, I would recommend splitting Fig 2 into multiple Figures.
4. There is a discrepancy between concentrations in graph 3A and the caption.
5. I would recommend moving Fig.3E and Fig. 4 to supporting information.
6. I would recommend showing results from Fig. 4 A-C graphically.
Author Response

(The authors gave the same response as above.)

Reviewer 3 Report
In this manuscript, the author reports, ‘Synthesis of a two-dimensional molybdenum disulfide nanosheet and ultrasensitive trapping of Staphylococcus aureus for enhanced photothermal and antibacterial wound-healing therapy’. The authors should address the following questions before getting a possible publication.
Recommendation: Major revisions needed as noted.
1. The novelty of the present work should be discussed in the Introduction section.
2. The formatting and grammatical errors in the article need to be checked carefully.
3. The author should write the purpose for each test in one/two sentences (in brief) before explaining the results of the characterization techniques. Therefore, the logic and organization of this part will be enhanced.
4. The scale bars in Fig. 7 are not clearly visible to the readers.
5. What does the error bars stand for in different Figures? It should be mentioned in the Figure captions
6. The authors have cited relevant references in the Introduction section; however the manuscript needs to be highlighted further to broaden the impact, related literatures: Journal of Materials Chemistry B, 7(15), 2534-2548; ACS Applied Materials & Interfaces, 12(46), 51940-51951; ACS Applied Materials & Interfaces, 12(26), 28952-28964; Applied Materials Today 20 (2020): 100781; Small, 15(12), 1900046; Ultrasonics Sonochemistry 60 (2020): 104797
Author Response

(The authors gave the same response as above.)

Round 2
Reviewer 3 Report
The manuscript can be accepted in the present form